# Did the COVID-19 Pandemic Prolong the Time Till Diagnosis and Worsen Outcomes for Children with Acute Appendicitis?

**DOI:** 10.3390/medicina57111234

**Published:** 2021-11-11

**Authors:** Idilė Vansevičienė, Danielė Bučinskaitė, Dalius Malcius, Aušra Lukošiūtė-Urbonienė, Mindaugas Beržanskis, Emilis Čekanauskas, Vidmantas Barauskas

**Affiliations:** 1Pediatric Surgery Department, Lithuanian University of Health Sciences, Eivenių Str. 2, LT-50161 Kaunas, Lithuania; dalius.malcius@lsmu.lt (D.M.); ausra.urboniene2@lsmu.lt (A.L.-U.); mindaugas.berzanskis@lsmu.lt (M.B.); emilis.cekanauskas@lsmu.lt (E.Č.); vidmantas.barauskas@lsmuni.lt (V.B.); 2Department of Surgery, Lithuanian University of Health Sciences, Eivenių Str. 2, LT-50161 Kaunas, Lithuania; daniele.bucinskaite@lsmuni.lt

**Keywords:** COVID-19, children, pediatric, surgery, appendicitis

## Abstract

*Background and Objectives*: Our aim was to see if the COVID-19 pandemic led to an increase of time until diagnosis, operation, and time spent in Emergency room (ER), and if it resulted in more cases of complicated appendicitis and complication rates in children. *Materials and Methods*: We conducted a retrospective analysis of patients admitted to the Pediatric Surgery Department with acute appendicitis during a 4-month period of the first COVID-19 pandemic and compared it to the previous year data—the same 4-month period in 2019. *Results*: During the pandemic, the time spent in the ER until arriving at the department increased significantly 2.85 vs. 0.98 h *p* < 0.001, and the time spent in the department until the operation 5.31 vs. 2.66 h, *p* = 0.03. However, the time from the beginning of symptoms till ER, operation time and the length of stay at the hospital, as well as the overall time until operation did not differ and did not result in an increase of complicated appendicitis cases or postoperative complications. *Conclusions*: The COVID-19-implemented quarantine led to an increase of the time from the emergency room to the operating room by 4 h. This delay did not result in a higher rate of complicated appendicitis and complication rates, allowing for surgery to be postponed to daytime hours if needed.

## 1. Introduction

Acute appendicitis (AA) remains one of the most common surgical problems worldwide in the pediatric population, having a lifetime risk of 7–8% [1]. Despite being a common surgical disease, its origin and cause remain unclear with many different speculations of possible lumen obstruction, changes in bacterial composition, genetic predisposition, and dietary and lifestyle choices being speculated in its pathogenesis [2,3,4,5,6,7]. Studies in immunology have shown different disease patterns in appendicitis forms and have provoked the thought of acute appendicitis having the possibility of either perforating or resolving, thus creating new possibilities of successful non-operative treatment with antibiotics [8,9]. However, the notion that complicated appendicitis must be operated on implements a no time-wasting strategy as there is more risk of perforation with time passed, thus no delay in presentation or diagnosis is crucial.

With more knowledge about the disease and changes in lifestyle, the disease, as any other, has begun to change. More so, over the recent year during the COVID-19 pandemic, the quantity of patients seeking help as well as the type of illness they present with has changed. Our hypothesis was that the pandemic increased the time until patients received surgical treatment, and that this resulted in an increase of complicated cases of appendicitis and postoperative complications.

## 2. Materials and Methods

We conducted a retrospective analysis of patients admitted to the Pediatric Surgery Department at the Hospital of Lithuanian University of Health Sciences during the first COVID-19 pandemic and nationwide quarantine—a 4-month period (from 16 March–16 June 2020—referred to as the pandemic group) and compared it to the previous year data, the same period of 4 months (from 16 March–16 June 2019—referred to as the non-pandemic group), selecting the patient records with the diagnosis of acute appendicitis, as diagnosed by the operating surgeon. The diagnosis of acute appendicitis was established using these criteria: pain in the right quadrant/lower abdomen/whole abdomen with or without pain migration; presence of fever >37.2 degrees Celsius, nausea or loss of appetite; presence of leukocytosis (elevated white blood cell count) >10 × 10 × 9/L, with neutrophilia >70% on blood tests; painful abdominal palpation on the right lower quadrant with muscle distention, with or without rebound tenderness; an inflamed appendix (diameter >6–7 mm) on ultrasound; or the presence of secondary appendicitis signs (free fluid, inflammation of surrounding tissue et cetera (etc.)). The type of appendicitis was decided by evaluating the intraoperative findings and histopathologic findings of the appendix. All patients with the diagnosis of acute appendicitis were operated on with preoperative antibiotic therapy and supportive treatment (analgesia, intravenous hydration, antipyretics, and antiemetics), as it is the choice of treatment for children with this diagnosis at our country and this hospital. Cases where patients were operated on with an unclear diagnosis, with possibility of acute appendicitis, but there were no pathological findings, or a different pathology was found—were not taken into the study due to the retrospective nature of the study and inability to identify all such cases, thus the negative appendectomy rate (NAE) was not evaluated.

All cases of acute appendicitis were categorized into types according to the operating surgeon’s diagnosis into uncomplicated and complicated appendicitis. Categorized as uncomplicated appendicitis: simple/catarrh—redness of the wall, dilation of appendiceal blood vessels; phlegmonous appendicitis—clear thickening of the appendix, presence of puss or fibrine on serous tissue without any possible gangrene or perforation present; and as complicated appendicitis: gangrenous appendicitis with the presence of fibrine and gangrene on any part of the appendiceal wall; perforated gangrenous—gangrene and a perforation seen, whether it is a minor perforation with clear signs of peritonitis with puss, feces etc. in the abdominal fluid, or a major perforation where the defect in the wall is clearly visible; also a periappendicular abscess was classified into this category, where the appendix is surrounded by an abscess with or without involvement of the omentum. The following data was analyzed: patient demographic data, duration of illness from onset of symptoms to arriving at the emergency room (ER); time spent from the ER to the surgical department and time passed from arrival to the department to the operating room (OR), type of appendicitis and postoperative complications, and length of stay at the hospital (pediatric surgery department and the pediatric intensive care unit). Because most physicians’ descriptions of duration of illness are highly unspecific, as well as patients or their parents being unable to pinpoint the exact time of the start of symptoms, these rules were applied to calculate the appropriate duration of illness (if not mentioned specifically): ill since morning/evening in accordance with arrival was taken as 12 h, ill since yesterday- as 24 h, 2 days as 48 h, a week as 120 h, etc. All the other time frames were measured from medical records as written. The clinical diagnosis was taken as the one that the patient was released with on medical records. All patients in 2020 received preoperative antibiotic treatment, intravenous hydration, and adequate analgesia upon establishing the diagnosis of possible acute appendicitis and were only operated on after receiving the COVID-19 virus polymerase chain reaction antigen test results. Until the virus antigen test results were received, patients were either isolated in the emergency room or at the department. No cases were COVID-19 positive during the time of this study. Statistical data analysis was performed by using IBM SPSS Statistics software. Quantitative data was found not to have a normal distribution using Kolmogorov and Smirnov tests; thus, a nonparametric Mann–Whitney test was used for the analysis. Results are described as median (minimum–maximum value). Qualitative data was compared using a Chi square for homogeneity criterion. Results are presented in absolute numbers and percentage. Obtained differences and relations were found as significant if *p* < 0.05.

## 3. Results

Three-hundred-and-seventy-five patients were admitted to the pediatric surgery department in 2020 from 16 March to 16 June during the first wave of COVID-19, 58 of which were treated for acute appendicitis. In comparison—a year prior in 2019 of the same period 16 March to 16 June, 815 patients were admitted, with 89 being treated for acute appendicitis.

A total number of 122 cases were analyzed. Due to various factors like incomplete length of symptoms (with patients being referred from other hospitals, uncertain diagnosis and prolonged ER visit, data glitches in the system) a few cases were dismissed, thus 52 cases in the year 2020 and 70 in 2019 were selected. There was a lower number of total patients admitted to the hospital in 2020 in comparison to 2019 (375 vs. 815), thus having a 54% lower number of patients admitted to the pediatric surgery department due to restrictions of elective surgery during the lockdown period. The number of acute appendicitis cases overall might have been lower, but the percentage of AA cases out of all admitted patients was slightly higher in 2020 (58 (15.47%) vs. 89 (10.92%)) due to a decrease of elective surgery cases. There was no significant difference in age (median 11.5 [3;17] vs. 11.5 [3;17] *p* = 0.546 or in distribution in age groups 0–7/8–13/14–17 (13.5%/46.2%/40.4% vs. 11.4%/60%/28.6%), *p* = 0.302.

When analyzing the type of appendicitis treated, we compared uncomplicated vs. complicated appendicitis cases, but there was no significant difference when comparing percentage of complicated cases in 2019 vs. 2020 (35.7% vs. 32.7%, *p* = 0,848) (Figure 1).

There was no significant difference between duration of symptoms from the beginning to ER (Table 1), from the beginning of symptoms to OR, the duration of operation, or the length of stay at the hospital. However, there was a significant increase in time spent in the Emergency room until ending up at the department (2.85 h [0.5;16,95] vs. 0.98 h [0.35;3], *p* < 0.001) and an increase in time passed from arriving at the department to arriving at the operating room (5.31 h [0.1;17.4] vs. 2.66 [0.17;20.07]).

## 4. Discussion

The beginning of the SARS-CoV-2 virus pandemic resulted in a nationwide quarantine and stay at home regime with a decline of regular social contact with other individuals, as well as closing of businesses, schools, and daycare centers. Hospitals were forced to rethink clinical practices, enforcing more preventive measures, more testing, and sustaining from performing elective surgery procedures while only focusing on emergency surgery.

The first wave of the pandemic was unexpected, and emphasis was put on the older population having a higher risk of disease and death, with children being pushed to the side and presented as not being affected by the virus or having little to no symptoms. However, in May 2020 in Wuhan, Cai et al., stressed to consider gastrointestinal symptoms in children, especially with fever and history of exposure to the virus [10]. Suwawongse in the US suggested that COVID-19 may present with various symptoms and may even present as possible appendicitis [11]. With more knowledge, a new pathology arose with similarities to Kawasaki disease, having heart, lung, skin, kidney, and gastrointestinal tract involvement, and according to Jackson et al., could even be mistaken for acute appendicitis [12].

Even though different regimes and tactics were employed in different countries with some having a stricter quarantine and others having a more lenient one, a general lessening of patients admitted to the ER was observed, possibly due to fear of contracting the virus at the hospital, with numbers rising when the restrictions were lifted [13,14]. We also observed a decrease of patients admitted to our department by 54%—from 815 the year prior to 375 during the first wave of the pandemic, and a lesser number of patients admitted for acute appendicitis from 89 to 58. A similar trend was also observed nationwide in the lesser number of performed appendectomies in children for the yearly statistics of 2020 and 2019 as presented by the Lithuanian Institute of Hygiene. It was also observed nationwide in Germany, UK, US, and Israel, with a decrease of 4–8 times being seen in Bosnia and Herzegovina [15,16,17,18,19,20,21]. We can clearly say that the majority of the number drop was due to elective surgery prohibition, which was one of the preventive measures issued by the government seeking to minimize possible spread of the pandemic. Other possible reasons for the decline could be social distancing, less possibility of contracting infectious diseases from others—with a change in bacterial composition and contraction of other pathogenetic bacteria from others having been discussed to play a role in the pathogenesis of acute appendicitis [2]—and the lessened factor of stress from having social interactions, school, separation from parents could also have played a role. Also, though the role of dietary specifics is only discussed as being a factor in acute appendicitis, it may also have contributed, as children stayed more at home and received more regular and possibly healthier options than they would have at school [3]. Self-medication could have also played a role with some cases of uncomplicated appendicitis resolving at home [22].

Due to the fact of self-medication, telemedicine, and fear of contracting the virus, we expected patients to seek help much later than the year prior and expected an increase of time from the beginning of symptoms to ER and OR as well as an increased number of perforations or complicated appendicitis cases due to a longer course of disease, as observed in most other countries, except single studies in China, the US, Turkey, and Italy, and more in children than adults [14,19,23,24,25,26,27,28,29,30,31,32]. We however did not observe a difference in complicated appendicitis rates—35.7% of all appendicitis cases in 2019 with a slightly, but not statistically significant, smaller number observed in 2020 during the pandemic of 32.7%. There were also no postoperative complications during the pandemic period, in comparison to two in the non-pandemic period, but it might be explained by no significant difference in duration of illness as we expected to see. This could be due to our early preoperative antibiotic and intravenous hydration therapy and analgesia, which were started as early as making the diagnosis and before reaching the operating room.

We also did not observe any difference in duration of illness from the onset of symptoms till seeking help at the emergency room, with the median time being 24 h in both groups, but we found a significant increase of time spent from arriving to the emergency room until the department by more than 1 h and 30 min and from the department to the operating room by more than 2 h and thirty minutes, which can be explained by isolation requirements until the COVID-19 antigen test results arrived, additional operating team preparations for the operation with adequate personal protective equipment (PPE), and other infection control measures employed by the staff. This led to a general increase in the median time spent from the onset of symptoms till arriving at the OR 33.54 h during the pandemic versus 27.77 h during the non-pandemic period—an increase of almost 6 h, but without statistical significance. We believe that this did have an impact on the patients as observed by a lower rate of perforations (although without significance), postoperative complications, and the same length of stay at the hospital, which was the same in both time periods—6 days. This was contradicting to many other reports online, where a higher duration of symptoms up to ER was observed both in adult patients and children, which may explain the higher rate of perforation and complicated appendicitis cases, as time is an important factor in the development of perforation; however, they do not measure the time from the ER to the department and to the OR [23,28,29,30,33,34,35,36]. Another factor that has importance in the development of complicated appendicitis and perforations could be overconfidence in the non-operative treatment method, which has been used more widely during the pandemic in some countries such as UK, Spain, and the US [35,37,38,39]. However it relies on predicting the type of appendicitis the patient has and is more used in uncomplicated cases, but since in most cases we are dealing with increased time until the patient receives the treatment, the risk of complicated appendicitis is higher and may require a longer operation with a larger incision as well as increasing the length of stay at the hospital [26,30,35]. Our hospital did not change its practice of operative treatment being the standard choice of treatment due to the pandemic, and it is possible that it might be another factor that led to no increase of complicated cases of acute appendicitis.

This leads us to believe that in our country, patients did not seek help later than before the pandemic, as we anticipated, and the time before the operation may have increased due to disease-preventing measures and extra testing for the COVID-19 virus. This unusual situation allows us to speculate that an increase of almost 4 to 6 h until receiving the operation with adequate preoperative antibiotic and supplementary treatment does not increase the risk of complicated appendicitis cases and early postoperative complications. We believe that it is possible without risk to the patient to postpone nighttime surgery to the next shift or daytime hours. It is of importance to the overworked surgeon as well as other staff members, as a smaller amount of sleep and exhausting shift work results in more mistakes and possible complications [40,41,42]. Other authors have supported this claim by saying that a short delay of 6 to even 11 h without signs of complicated appendicitis does not result in a more advanced disease, length of hospital stay, or postoperative complications, with emphasis being put on early antibiotic therapy; however, they significantly increase when that time exceeds 18–24 h [37,43,44,45,46]. No difference was also found when nighttime surgery was postponed to daytime surgery in children but was advised against in adults [47,48,49].

## 5. Conclusions

COVID-19-implemented quarantine did not have any significant delay in patients seeking help for acute appendicitis. Due to newly implemented pandemic measures, patients spent almost 4 h longer between arriving at the ER to ending up in the operating room. In our study, this delay did not result in higher rates of complicated appendicitis or postoperative complication rates, nor did it increase the length of stay at the hospital; thus, surgery could be postponed from nighttime to daytime hours without risk to the patient if early antibiotic and supplementary treatment is given.

## Figures and Tables

**Figure 1 medicina-57-01234-f001:**
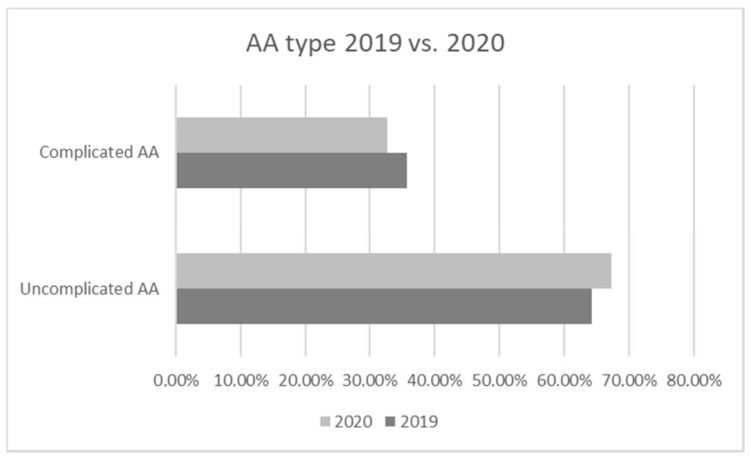
The rate of uncomplicated vs. complicated acute appendicitis(AA) during the pandemic (2020) and non-pandemic (2019) period.

**Table 1 medicina-57-01234-t001:** Comparison of different time intervals from the beginning of symptoms till arriving at the emergency room, the operating room, and time spent in each department, as well as duration of operation.

Duration	2019/Non-Pandemic	2020/Pandemic	Significance
Duration of symptoms from start to ER ^1^	24 [4;120]	24 [3;120]	*p* = 0.658
Time from ER to Department ^1^	0.98 [0.35;3]	2.85 [0.5;16.95]	*p* < 0.001 *
Time from Department to OR ^1^	2.66 [0.17;20.07]	5.31 [0.1;17.4]	*p* = 0.03 *
Time from the beginning of symptoms to OR ^1^	27.77 [6.58;122.83]	33.54 [12.87;144.17]	*p* = 0.095
Duration of operation ^1^	1.1 [0,5;3]	1 [0.5;3.75]	*p* = 0.749
Length of stay at the hospital ^2^	6 [1;20]	6 [2;14]	*p* = 0.074

^1^ data presented in hours; ^2^ data presented in days; all intervals reported as median [minimum; maximum]; * if significant, marked.

## Data Availability

The data presented in this study are available on request from the corresponding author.

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
