# Peer review of "Did the COVID-19 Pandemic Prolong the Time Till Diagnosis and Worsen Outcomes for Children with Acute Appendicitis?"

_medicina, 2021, doi:10.3390/medicina57111234_

Round 1

Reviewer 1 Report

Thank you authors for this work. Appendicitis is the bread and butter of general surgery. Like the impact of COVID pandemic is felt throughout all surgical practice, especially during first wave of the pandemic.

These results are similar and in line with other international studies which present similar impact on surgical practice.

Interesting to see that there was no delayed presentations with acute appendicitis, which was an issue in other countries.
Can the authors please discuss any possible explanation as to why Lithuanian patients did not have delayed presentations compared to their international counterparts?
In our practice , we found that most of the delays were due to additional COVID tests that are necessary prior to operations, and also the impact of PPE, isolation requirements and infection control measures in theatres for these patients, did the authors experience similar delays?

Can the authors elaborate in their results whether an increase in the rate of complicated appendicitis was noticed as a result of these delays ? 
Was there increased utilisation of non operative management of appendicitis during pandemic period? Did the authors notice increased use of radiology in the diagnosis of appendicitis preoperatively? 

Author Response

  • Can the authors please discuss any possible explanation as to why Lithuanian patients did not have delayed presentation compared to their international counterparts?

We expected later presentations, that was one of the reasons why the study was done, however we indeed found that the median time from the beginning of symptoms to arriving at the ER was the same 24 hours. It could be due to the change of family medicine practices in Lithuania. Most practices were closed during lockdown and operating mostly long distance, by giving consultations via telephone. In most cases as we have noticed parents either skipped the call altogether and ended up at the ER themselves or if they did call the family medicine doctor, the doctor himself did not see the patient or perform tests on him, but rather would always suggest going to the ER for a checkup and testing. Thus all patients came directly from home to the ER and all the tests were performed there as well as consultations by pediatricians and pediatric surgeons. (This has been added to the revised manuscript)

  • In our practice, we found that most of the delays were due to additional COVID tests that are necessary prior to operations, and also the impact of PPE, isolation requirements and infection control measures in theatres for these patients, did the authors experience similar delays?

We also experienced delays from Emergency room till the department (median 0.98 h. vs. 2.85 h) and from arriving to the department until the operating room (median 2.66 h vs. 5.31 h) and we strongly believe that is was also due to testing for covid-19 before the operation (we did not measure the amount of time from doing the test until the test results came in, however in our general experience it is about 3-5 hours), isolation requirements (many times patients were isolated in the ER or in the department, depending on the situation) using PPE. (also added to the revised manuscript).

  • Can the authors elaborate in their results whether an increase in the rate of complicated appendicitis was noticed as a result of these delays?

Due to request from both reviewers, we have changed our classification of appendicitis into complicated and uncomplicated and have discussed this in the revised manuscript. We found no significant increase in rate of complicated appendicitis due to those delays, which allowed us to draw the conclusion that this amount of delay does not heighten the chances of complicated appendicitis cases and allows for surgery to be pushed if need be.

  • Was there increased utilization of non-operative management of appendicitis during pandemic period? Did the authors notice increased use of radiology in the diagnosis of appendicitis preoperatively?

The pandemic did not increase percentage of NOM as it is not yet common practice in children with acute appendicitis in our country. We have read reports of it being used in cases of suspected uncomplicated appendicitis with various degrees of success, but it is yet to become common practice in Lithuania. Our gold standard of treatment is operative, with preoperative antibiotic therapy and supportive treatment. However, we are working towards the goal of understanding the possibilities where using NOM is safe and when it should not be used or reverted to surgery instead. This is one of those studies that help us understand the importance of duration of the disease for outcomes of acute appendicitis and we are also working on understanding other aspects as well (we are currently performing the study). The changes in radiology were not taken into account here as in Lithuania the gold standard of radiological evaluation remains ultrasound, which is performed for all children with abdominal pain, who enter the emergency room. There have been articles about increased use of CT to diagnose appendicitis in the US, however it is not common in Lithuania as we try to limit the amount of radiation a child receives, and in only very rare instances is it used. No patients in this current study had a CT performed, thus we did not mention it altogether.

Thank you very much for your revisions and advice on improvement, we sincerely hope that we were able to correct any unclear and incomplete parts of the manuscript.

Reviewer 2 Report

Dear authors

I would like to congratulate you for presenting a well-constructed study. This is the 'most common research topic in the present times'. The manuscript is interesting and has merit. However, I have several comments that need to be addressed.

Abstract: well-written. No changes are needed.

Introduction: Please write your hypothesis at the end of the Introduction section.

Methods

- What are the diagnostic criteria of acute appendicitis at your center?

-What is meant by type of appendicitis? Do you mean complicated and uncomplicated? Please elaborate on the operational definitions which you use at your center?

-Which antigen test is used for COVID-19 before the surgery? Were any cases that were COVID-positive? Please mention.

Results:

-Did you operate upon all the cases of AA or some of them were managed via the non-operative approach (NOM)?

-Do you label perforated vs non-perforated based on the operative findings or histopathology (HPE)? Please mention.

-Were there any cases where HPE (negative appendectomy) was normal?

-I would have used complicated and non-complicated appendicitis rather than perforated vs non-perforated. It is a broader definition that also includes phlegmon, complex abscess, gangrenous appendix. Can you mention these?

Discussion:

It must be mentioned that there are two factors that increase the rates of perforation: an increase in the duration of symptoms (onset to ER) and over-reliance on NOM. Your finding of no difference in the perforation rates is because you had not managed any patients via the NOM approach. Please discuss this in the discussion section.

Author Response

  • Introduction: Please write your hypothesis at the end of the Introduction section.

Our hypothesis was that the Covid-19 pandemic increased the time until patients received surgical treatment and it resulted in increase of complicated cases of appendicitis and postoperative complications. (Added to the revised manuscript)

  • Methods: What are the diagnostic criteria of acute appendicitis at your center?

Acute appendicitis is diagnosed by combining patient history: pain in the right quadrant/lower abdomen/whole abdomen with or without pain migration; Presence of fever >37,2 degrees Celsius, Nausea or poor eating, presence of leukocytosis (elevated blood cell count) >10x10*9/l, with neutrophilia >70%, painful abdominal palpation on the right lower quadrant with muscle distention, rebound tenderness, an inflamed appendix (diameter >6-7 mm) on ultrasound, or presence of secondary appendicitis signs (free fluid, inflammation of surrounding tissue etc.). However not all signs may be present, and the diagnosis is made by the pediatric surgeon on call.  (We have added them to the revised manuscript).

  • What is meant by type of appendicitis? Do you mean complicated and uncomplicated? Please elaborate on the operational definitions which you use at your center?

By the type of appendicitis, we meant perforated and unperforated, but due to notice from both reviewers have decided that it would be better to categorize into uncomplicated and complicated appendicitis. Uncomplicated appendicitis: simple/catarrh- redness of the wall, dilation of appendiceal blood vessels, phlegmonous appendicitis clear thickening of the appendix, presence of puss, fibrine on serous tissue- all without any possible gangrene or perforation present; and complicated appendicitis: gangrenous appendicitis- presence of fibrine and gangrene on any part of the appendiceal wall; perforated gangrenous- gangrene and a perforation seen- whether it is a minor perforation with clear signs of peritonitis- with puss in abdominal fluid, feces and etc. or a bigger perforation where the defect in the wall is clearly visible, periappendiceal abscess- where the appendix is surrounded by an abscess with or without involvement of the omentum. This classification was inspired by M.Ruber’s and M.Minderjahn’s work on immune studies in the appendix, showing a different cytokine pattern and inflammatory cell pattern in uncomplicated and complicated appendicitis. (we have changed the classification in the revised manuscript).

  • Which antigen test is used for Covid-19 before the surgery? Were any cases that were covid-positive? Please mention.

Before surgery all patients undergo the Polimerase Chain Reaction Covid-19 ELISA test. No cases were covid-19 positive during the time of this study. (This has also been added to the revised manuscript)

  • Results: Did you operate upon all cases of AA or some of them were managed via the non-operative approach (NOM)?

It is still the gold standard for children in Lithuania to be treated with operation with a preoperative antibiotic prophylactic and supportive care (intravenous hydration, analgesia, antipyretics, antiemetics and so on.) as clear guidelines when NOM is safe to use on children have yet to be established in our country. We are working on more follow up studies to understand this pathology better to improve treatment and possibly understand when it is safe to use NOM and when it should not be attempted.

  • Do you label perforated vs. non-perforated based on the operative findings or histopathology (HPE)? Please mention.

The labeling was done according to the operating surgeon’s diagnosis along with the histopathology report. (added to the revised Manuscript)

  • Were there any cases where HPE (negative appendectomy) was normal?

Cases where patients were operated on with an unclear diagnosis, with possibility of acute appendicitis, but there were no pathological findings, or a different pathology was found were not taken into the study due to the retrospective nature of the study and inability to identify all such cases. ( added to the revised Manuscript).

  • I would have used complicated and non-complicated appendicitis rather than performed vs. non-performed. It is a broader definition that also includes phlegmon, complex abscess, gangrenous appendix. Can you mention these?

We have addressed the issue and changed the analysis and the description in the advised manuscript, and I believe I have mentioned this in one of the questions above, thank you very much for pointing this out to us. This classification is probably much better in comparison.

  • Discussion: It must be mentioned that there are two factors that increase the rates of perforation: an increase in the duration of symptoms (onset to ER) and over-reliance on NOM. Your finding of no difference in the perforation rates is because you had not managed any patients via the NOM approach. Please discuss this in the discussion section.

We have addressed the importance of time as a factor as it had been thought to be one of the most important factors on stage of acute appendicitis, however studies with immunology (Ruber et al, Minderjahn et al), have shown that there is possibility that uncomplicated appendicitis might have a different immune mechanism and thus might not progress to gangrene and perforation over time. Our study could support this notion, as with increased time we had similar results of complicated appendicitis during the pandemic as before. We plan on analyzing more data on this subject in our later studies. The over reliance on NOM could be one of those factors that lead to complicated cases, as there is no 100 percent sure way to tell while using NOM if an uncomplicated or complicated appendicitis case is being treated. That is one of the reasons that we do not yet use the NOM approach until we have more thorough data on when it is safe to use on children. We tried to add this to our discussion section as suggested. ( We have added your suggested points to the discussion).

Thank you very much for your revisions and advice on improvement, we sincerely hope that we were able to correct any unclear and incomplete parts of the manuscript.

Round 2

Reviewer 2 Report

The authors have addressed my comments. The overall scientific quality has improved significantly. The work has merit and will be of interest to our readers. I congratulate the authors for their work. Only, a spell check is needed for minor grammatical errors.